# Horospherical Decision Boundaries for Large Margin Classification in Hyperbolic Space

**Xiran Fan**
Department of Statistics
University of Florida
fanxiran@ufl.edu

**Chun-Hao Yang**
Institute of Statistics and Data Science
National Taiwan University
chunhaoy@ntu.edu.tw

**Baba C. Vemuri**
Department of CISE
University of Florida vemuri@ufl.edu

## Abstract

Hyperbolic spaces have been quite popular in the recent past for representing hierarchically organized data. Further, several classification algorithms for data in these spaces have been proposed in the literature. These algorithms mainly use either hyperplanes or geodesics for decision boundaries in a large margin classifiers setting leading to a non-convex optimization problem. In this paper, we propose a novel large margin classifier based on horospherical decision boundaries that leads to a geodesically convex optimization problem that can be optimized using any Riemannian gradient descent technique guaranteeing a globally optimal solution. We present several experiments depicting the competitive performance of our classifier in comparison to SOTA.

## 1   Introduction

Hyperbolic space, a non-Euclidean space with constant negative curvature, has been shown [25, 23, 29, 24] to be effective for representing hierarchically organized data. For example, authors in [25] showed that a tree can be embedded in a hyperbolic space with arbitrarily small distortion. The main reason for this is that a hyperbolic space can be regarded as a continuous version of trees – the volume of the space grows *exponentially* as one moves away from the center in hyperbolic space. This matches the growth pattern of the number of nodes in a tree which grows *exponentially* as the depth of the tree increases. Hyperbolic space embedding has been shown to be a promising approach for representing data with a (latent) hierarchical structure [23, 24, 29, 15].

Recently, representation of data in hyperbolic space for the fundamental tasks of unsupervised and supervised learning has been popularized in various contexts, e.g., dimensionality reduction [10, 13], clustering [22], large-margin classifier [12, 32, 11], regression [20], etc. Existing 'linear' classifiers in hyperbolic spaces are predominantly based on *geodesics* i.e., using geodesics as decision boundaries. In [12], the decision boundary is chosen to be the intersection of the hyperboloid model and a hyperplane in the ambient space, which in this case is the Minkowski space. Then, the support vector machine (SVM) in hyperbolic space is formulated as a nonconvex optimization problem. In [32], authors followed the same parameterization of the hyperbolic geodesic decision plane and provided a series of algorithms to provably learn large margin classifiers in hyperbolic space. However, as pointed out by [11], the algorithm in [32] fails to converge in practice. Authors in [11] used the Poincaré ball model and parameterized the geodesic decision plane as a hyperplane mapped using the exponential map from the *tangent space* at some reference point. They first constructed convex hulls for each data cluster in the hyperbolic space and the reference point is then chosen to be the midpoint

between different convex hulls. Then they apply a *Euclidean* perceptron/SVM algorithm to data lifted into the tangent space at the aforementioned reference point. Although the optimization problem in the tangent space is convex, the procedure of the tangent space approximation introduces inaccuracies and distortions. Moreover, convex hull learning is highly unstable and their implementation is only applicable to the 2-dimensional hyperbolic space.

Finally, it is worth mentioning that linear classification within hyperbolic space, which can be considered as the last layer, referred to as the hyperbolic logistic regression (LR), is a fundamental component of hyperbolic neural networks (HNNs) [16, 27]. The calculation of logits in this layer is based on the distances between samples and the geodesic decision boundary. Notably, this hyperbolic LR employs a geodesic decision boundary but is not a large-margin classifier.

## 1.1 Horospherical Decision Boundaries for Classification in Hyperbolic Sapce

*Horospheres*, which are the level sets of the *Busemann function* in hyperbolic spaces, are the analogs of Euclidean hyperplanes [4]. Horospheres (horocycles) are contained in the Poincaré ball (disk) and are tangential to the ball (disk) at an ideal point as shown in Figure 1. A collection of horospheres centered at the same ideal point are parallel to each other and the lengths of geodesic segments between two horospheres are all equal, just as the lengths of line segments between parallel hyperplanes in Euclidean space are all equal. This property of horospheres was explored by [9] to develop a dimensionality reduction method for data in hyperbolic space. By using horospherical projection, they are able to preserve the distance information in the original data. However, there is no literature on constructing a 'linear' classifier in hyperbolic space using horospheres as the decision boundaries although the horospheres are the hyperbolic equivalent of Euclidean hyperplanes. Therefore, it is natural to consider the use of horospheres as decision boundaries for classifi-

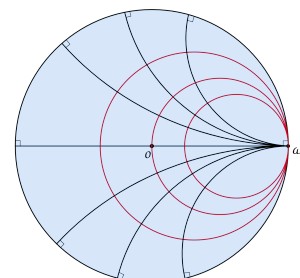

Figure 1: A 2-d Poincaré disk model $\mathbb{B}^2$ and its boundary $\partial\mathbb{B}^2 = \mathbb{S}^1$. Given an ideal point $\omega \in \partial\mathbb{B}^2$, the black lines and curves are hyperbolic geodesics starting (ending) at $\omega$ and the red circles are horocycles centered at $\omega$.

cation in hyperbolic spaces. In this work, we propose a novel *hyperbolic large-margin classifier using horospheres as decision boundaries in the Poincaré model*. We term this classifier as a *HoroSVM*. The horospheres are well-defined in the Poincaré ball model. A toy example as shown in Figure 2 demonstrates the advantage of horospherical decision boundaries over geodesic decision boundaries. As the tree-structured data grows in depth, leaf nodes are embedded closer to each other within a subtree and among different subtrees. One of the classification problems in hyperbolic space is to determine whether a node belongs to a chosen subtree given the embedding. For comparison purposes, the decision boundaries of HoroSVM (Figure 2(a)) and hyperboloid SVM [12] (Figure 2(b)) are shown in the figure. As evident, the horospherical decision boundary perfectly separates (the root node is excluded in training) the data while the geodesic decision boundary makes several mistakes on both positive and negative samples. We present a novel formulation of the classification problem in the hyperbolic space as a geodesically convex optimization problem on a Riemannian manifold. This optimization problem can be easily solved using any Riemannian gradient descent technique guaranteeing global optimality. Gradient-based optimizations for geodesically convex problems guaranteeing global optimal solutions are the topic of investigation in optimization literature and we refer the reader to [35] for detailed convergence analysis of several such optimization methods. Further, we empirically validate our method on several real and synthetic data sets.

It should be noted that a horosphere decision boundary has been used in some recent works [31, 28] in constructing HNNs. For example, authors in [31] proposed hyperbolic neuron models using the Busemann function as a generalization of the Euclidean inner product to extract horosphere features from data. Authors in [28] proposed a shallow fully-connected continuous network spanned by (hyperbolic) neurons, on noncompact symmetric space (including hyperbolic space) using the Helgason-Fourier transform. Authors in [33] produced Euclidean features from hyperbolic embeddings via the eigenfunctions of the Laplace operator in the hyperbolic space where the eigenfunctions involve horosphere features. Note that none of the above works developed a large margin classifier using the horosphere as a decision boundary. To the best of our knowledge, our work is the first in the

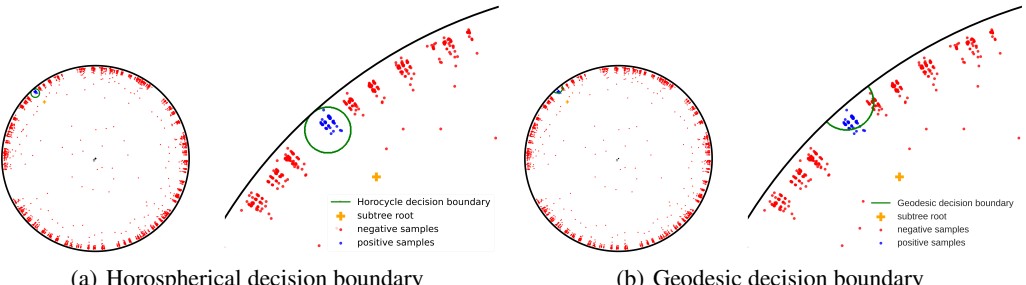

(a) Horospherical decision boundary        (b) Geodesic decision boundary

Figure 2: A balanced tree with depth 6 and spread 4 embedded in a 2-d Poincaré ball model using [15] is depicted in the figure. The orange plus node is the root of a chosen subtree, the blue dots are positive samples (nodes of the subtree) and the red dots are negative samples. (a) depicts a HoroSVM performance on the classification of positive and negative samples/nodes along with a zoomed-in version on its right. (b) depicts the geodesic boundary from the competing method, Hyperboloid SVM [12], along with the zoomed-in version to its right.

literature to present a convex optimization formulation of a large-margin classifier using a horosphere decision boundary in a hyperbolic space.

The rest of this paper is organized as follows. In Section 2, we present some background on hyperbolic geometry pertinent to the work presented here. In Section 3, we present our horospherical boundary-based classification methods. Experimental results are presented in Section 4 to demonstrate the advantage of our HoroSVM over competing hyperbolic classifiers. Finally, we conclude in Section 5.

## 2 Background

In this section, we review some basic concepts of hyperbolic geometry including the generalization of the Euclidean hyperplane to the hyperbolic space namely, the horosphere.

### 2.1 Hyperbolic Space and the Poincaré Ball Model

There are five isometric models of the hyperbolic space: the Poincaré ball model, the Lorentz model, the Klein model, the upper-half space model, and the Hemisphere model [7]. We choose the Poincaré Ball model in this paper as it is easy to visualize and the Busemann function has a nice closed-form expression in this model. Note that the decision boundary of choice in our work is the level-set of the Busemann function namely, the horosphere.

An $n$-dimensional Poincaré Ball model, denoted by $(\mathbb{B}^n, g_{\mathbb{B}})$, consists of all points in an open ball of radius 1, i.e., $\mathbb{B}^n = \{\boldsymbol{x} \in \mathbb{R}^n : \|\boldsymbol{x}\| < 1\}$, and equipped with the Riemannian metric $g_{\mathbb{B}}(\boldsymbol{x}) = 4(1 - \|\boldsymbol{x}\|^2)^{-2} g_{\mathbb{R}}$, where $\|\cdot\|$ is the Euclidean $L_2$ norm, and $g_{\mathbb{R}}$ is the Euclidean metric. The geodesic distance between points $\boldsymbol{x}, \boldsymbol{y} \in \mathbb{B}^n$ is $d_{\mathbb{B}}(\boldsymbol{x}, \boldsymbol{y}) = \cosh^{-1}\left(1 + 2\frac{\|\boldsymbol{x}-\boldsymbol{y}\|^2}{(1-\|\boldsymbol{x}\|^2)(1-\|\boldsymbol{y}\|^2)}\right)$.

### 2.2 Horospheres

**Geodesics, geodesic rays, and ideal points** The shortest path that connects two points in the Poincaré Ball model is called a *geodesic segment*. A *geodesic ray* is a geodesic segment that can be infinitely extended in one direction. We call the endpoint at infinity of a geodesic ray an *ideal point*. For $\mathbb{B}^n$, ideal points form the boundary of the ball: $\partial\mathbb{B}^n = \mathbb{S}^{n-1} = \{x \in \mathbb{R}^n : \|\boldsymbol{x}\| = 1\}$, where $\mathbb{S}^{n-1}$ is the $(n-1)$-dimensional hypersphere. The hypersphere $(\mathbb{S}^{n-1}, g_{\mathbb{S}})$ is a Riemannian manifold equipped with the Riemannian metric $g_{\mathbb{S}} = 4(1 + \|\boldsymbol{x}\|^2)^{-2} g_{\mathbb{R}}$.

**Busemann function [6]** Let $\boldsymbol{\omega} \in \partial\mathbb{B}^n$ be an ideal point and $\gamma_{\boldsymbol{\omega}} : [0, \infty) \to \mathbb{B}^n$ a geodesic ray pointing $\boldsymbol{\omega}$. The *Busemann function* is defined as

$$b_{\boldsymbol{\omega}}(\boldsymbol{x}) = \lim_{t \to \infty} (d(\gamma_{\boldsymbol{\omega}}(t), \boldsymbol{x}) - t), \quad \boldsymbol{x} \in \mathbb{B}^n. \tag{1}$$

In the Poincaré Ball model, Eq. (1) has a closed form : $b_{\boldsymbol{\omega}}(\boldsymbol{x}) = -\log \frac{1-\|\boldsymbol{x}\|^2}{\|\boldsymbol{\omega}-\boldsymbol{x}\|^2}$.

**Horospheres [6]**   In $\mathbb{B}^n$, a horosphere is a $(n-1)$-dimensional sphere that is internally tangent to $\partial\mathbb{B}^n$ at an ideal point. For a given $\boldsymbol{\omega} \in \partial\mathbb{B}^n$, the level sets of Busemann function $b_{\boldsymbol{\omega}}(\boldsymbol{x})$ in the Poincaré ball model is a series of horospheres tangent at $\boldsymbol{\omega}$. Horospheres are hyperbolic hyperplanes in the sense that the corresponding construction in Euclidean space gives a hyperplane.

Given an ideal point $\omega \in \partial\mathbb{B}^n$, the function defined by $\langle \omega, x \rangle_B : x \mapsto -b_{\omega}(x)$ is constant over horosphere tangent at $\omega$.[1] Hence any horosphere can be parameterized with an ideal point $\omega$ and an offset value $b$ of the level set.

Let $\Pi$ denote the set of horospheres of $\mathbb{B}^n$. A horosphere $\pi \in \Pi$ can be parameterized by $0 < \mu \in \mathbb{R}^+$ [2], $\boldsymbol{\omega} \in \mathbb{S}^{n-1}$, and $b \in \mathbb{R}$ as

$$\pi_{\mu,\boldsymbol{\omega},b} := \{\boldsymbol{z} \in \mathbb{B}^n | \mu\langle \boldsymbol{\omega}, \boldsymbol{z} \rangle_{\mathbb{B}} - b = 0\}. \tag{2}$$

We will use $\pi$ or $\pi_.$ to represent a horosphere in different parameterizations hereafter.

## 3   Horospherical Boundary-based Classification

In this section, we present the key theoretical contributions of our work namely, a horosphere-based SVM classifier that involves formulating and solving a geodesically convex optimization problem. First, we present some preliminary facts and results about the horospheres. Then, we present the optimization problems for the horospherical perceptron and SVM along with analysis.

### 3.1   Point to Horosphere Distance

While the distance from the origin $\boldsymbol{o}$ to a horosphere has been known for decades [18] (Introduction 4.1, p.31), we present a natural generalization of previous results by providing a closed-form expression for measuring the hyperbolic distance from any arbitrary point $\boldsymbol{x} \in \mathbb{B}^n$ to a given horosphere $\pi_{\mu,\boldsymbol{\omega},b}$. The following remark provides this result.

**Proposition 3.1.** *Let $\pi_{\mu,\boldsymbol{\omega},b}$ be a horosphere. The hyperbolic distance of a point $\boldsymbol{x} \in \mathbb{B}^n$ to a horosphere $\pi_{\mu,\boldsymbol{\omega},b}$ is given by*

$$d_{\mathbb{B}}(\boldsymbol{x}, \pi_{\mu,\boldsymbol{\omega},b}) = \frac{|\mu\langle \boldsymbol{\omega}, \boldsymbol{x} \rangle_{\mathbb{B}} - b|}{\mu}. \tag{3}$$

Notice that it shares a similarity to the Euclidean distance of a point to a hyperplane. Before presenting the proof for Proposition 3.1, we recall the following Fact 3.2 and Lemma 3.3 from [31].

**Fact 3.2.** *Given an ideal point $\boldsymbol{\omega}$ and a point $\boldsymbol{x} \in \mathbb{B}^n$, there is a unique horosphere passing through $\boldsymbol{x}$ and tangent at $\boldsymbol{\omega}$.*

**Lemma 3.3.** *[31] Let $\Pi_{\boldsymbol{\omega}}$ be the set of horocycles of $\mathbb{B}^n$ tangent at $\boldsymbol{\omega}$. Given $\lambda \in \mathbb{R}$, let $\pi_{\lambda,\boldsymbol{\omega}}$ be the unique horosphere that passes through $\tanh(\lambda/2)\cdot\boldsymbol{\omega}$ and tangent at $\boldsymbol{\omega}$. Note that $\Pi_{\boldsymbol{\omega}} = \cup_{\lambda\in\mathbb{R}}\{\pi_{\lambda,\boldsymbol{\omega}}\}$. We have the following two results: (i) the hyperbolic lengths of geodesic (that pass through $\boldsymbol{\omega}$) segments between $\pi_{\lambda_1,\boldsymbol{\omega}}$ and $\pi_{\lambda_2,\boldsymbol{\omega}}$ are equal to $|\lambda_1 - \lambda_2|$; (ii) $\langle \boldsymbol{\omega}, x \rangle_{\mathbb{B}} = \lambda$ for any $\boldsymbol{x} \in \pi_{\lambda,\boldsymbol{\omega}}$.*

Figure 3 shows a 2D Poincaré disk model $\mathbb{B}^2$ and its boundary $\mathbb{S}^1$. The point $o$ is the origin of the disk, $\boldsymbol{x} \in \mathbb{B}^2$ is a point, and $\boldsymbol{\omega} \in \mathbb{S}^1$ is a point at infinity (an ideal point). Two geodesics ending at the same $\boldsymbol{\omega}$ from $\boldsymbol{x}$ and $o$ respectively are shown in the figure (black solid line/curve). The circle $\pi_{\mu,\boldsymbol{\omega},b}$ is a given horocycle tangent (red solid circle) at $\boldsymbol{\omega}$. The hyperbolic distance $d_{\mathbb{B}}(\boldsymbol{x}, \pi_{\mu,\boldsymbol{\omega},b})$ between $\boldsymbol{x}$ and $\pi_{\mu,\boldsymbol{\omega},b}$ is identified as the distance between $\boldsymbol{x}$ and $\boldsymbol{y}_{\boldsymbol{x}}$ where $\boldsymbol{y}_{\boldsymbol{x}}$ is the projection of $\boldsymbol{x}$ to $\pi_{\mu,\boldsymbol{\omega},b}$ along the geodesic ending at $\boldsymbol{\omega}$. Let $\pi^{\boldsymbol{x}}$ (red dashed circle) be the unique horocycle that passes through $\boldsymbol{x}$ and is tangent at $\boldsymbol{\omega}$. Note that the lengths of all geodesic segments between two horocycles are the same. That is, $d_{\mathbb{B}}(\boldsymbol{x}, \pi_{\mu,\boldsymbol{\omega},b}) = d_{\mathbb{B}}(\boldsymbol{x}, \boldsymbol{y}_{\boldsymbol{x}}) = d_{\mathbb{B}}(\boldsymbol{x}_0, \boldsymbol{y}_0)$, where $\boldsymbol{x}_0, \boldsymbol{y}_0$ are *horocyclic projections* [9] of $\boldsymbol{x}, \boldsymbol{y}_{\boldsymbol{x}}$ along $\pi^{\boldsymbol{x}}$ and $\pi_{\mu,\boldsymbol{\omega},b}$ respectively.

---

[1]Note that the Euclidean inner product $\langle \boldsymbol{w}, \cdot \rangle$ is constant over a hyperplane that is perpendicular to a given direction $\boldsymbol{w}$.

[2]Parameter $\mu$ is included for simplicity in analysis later on (Eq 3, 12)

*Proof of Proposition 3.1.* Given a point $\boldsymbol{x} \in \mathbb{B}^n$ and an ideal point $\boldsymbol{\omega} \in \mathbb{S}^{n-1}$, let $\pi \in \Pi_{\boldsymbol{\omega}}$ be a horosphere tangent at $\boldsymbol{\omega}$ and $\pi^{\boldsymbol{x}} \in \Pi_{\boldsymbol{\omega}}$ be the unique horosphere that passes through $\boldsymbol{x}$. Write $\langle \boldsymbol{\omega}, \boldsymbol{x} \rangle_{\mathbb{B}} = \lambda_{\boldsymbol{x}}$ and then $\boldsymbol{x}_0 = \tanh(\lambda_{\boldsymbol{x}}/2) \cdot \boldsymbol{\omega}$ is the horospherical projection of $\boldsymbol{x}$ along $\pi_{\boldsymbol{x}}$. For consistency in notation, we write $\pi^{\boldsymbol{x}}$ as $\pi_{\lambda_{\boldsymbol{x}}, \boldsymbol{\omega}}$. Since the horosphere $\pi_{\mu, \boldsymbol{\omega}, b}$ can be reparameterized as $\pi_{\lambda, \boldsymbol{\omega}}$, where $\lambda = \frac{b}{\mu}$, the hyperbolic distance from $\boldsymbol{x}$ to $\pi_{\mu, \boldsymbol{\omega}, b}$ is the hyperbolic distance between $\pi_{\lambda_{\boldsymbol{x}}, \boldsymbol{\omega}}$ and $\pi_{\lambda, \boldsymbol{\omega}}$ which is $|\lambda_{\boldsymbol{x}} - \lambda| = \frac{|\mu \langle \boldsymbol{\omega}, \boldsymbol{x} \rangle_{\mathbb{B}} - b|}{\mu}$ (by Lemma 3.3). This completes the proof. ∎

## 3.2 Horospherical Decision Boundaries

We consider classification problems in hyperbolic space of the following form: $\mathcal{X} \subset \mathbb{B}^n$ denotes the feature space and $\mathcal{Y} = \{\pm 1\}$ denotes the binary label space. In the following, we denote the training set by $S \subset \mathcal{X} \times \mathcal{Y}$. The decision rule using a horosphere as its decision boundary can be written as the following function $f : \mathcal{X} \mapsto \mathcal{Y}$ where

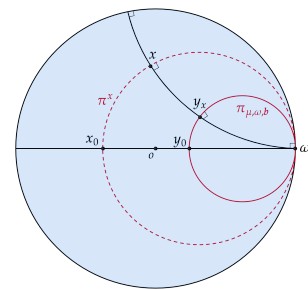

$$f(\boldsymbol{x}; \mu, \boldsymbol{\omega}, b) = \text{sign}\left(\mu \langle \boldsymbol{\omega}, \boldsymbol{x} \rangle_{\mathbb{B}} - b\right). \qquad (4)$$

The positive samples are expected to lie inside a horosphere while the negative samples are expected to lie outside a horosphere. This is analogous to the linear decision boundary in Euclidean space and we will build a horospherical perceptron and a horospherical SVM based on this decision boundary.

Figure 3: The relationship between horocycles, geodesic, and the horocyclic projections in $\mathbb{B}^2$.

It should be noted that in $\mathbb{R}^n$ the hyperplane $\xi_{a, \boldsymbol{w}, b} = \{\boldsymbol{z} \in \mathbb{R}^n | a \langle \boldsymbol{w}, \boldsymbol{z} \rangle - b = 0\}$ where $a \in \mathbb{R}^+, b \in \mathbb{R}$, $\boldsymbol{w} \in \mathbb{S}^{n-1}$ is the hyperplane $\xi_{a, -\boldsymbol{w}, -b}$. However, the horospheres $\pi_{\mu, \boldsymbol{\omega}, b}$ and $\pi_{\mu, -\boldsymbol{\omega}, -b}$ respectively represent two distinct horospheres, centered at $\boldsymbol{\omega}$ and $-\boldsymbol{\omega}$ respectively. Let $\Pi^+ = \{\pi_{\mu, \boldsymbol{\omega}, b} \in \Pi | b > 0\}$ and $\Pi^- = \{\pi_{\mu, \boldsymbol{\omega}, b} \in \Pi | b < 0\}$. Thus, $\Pi^+, \Pi^- \subset \Pi$ and the radius of $\pi \in \Pi^+$ is less than $1/2$ and the radius of $\pi \in \Pi^-$ is greater than $1/2$. In most cases of classification in hyperbolic space, the positive samples are clustered near the boundaries. Hence, we restrict ourselves to finding a horosphere $\pi \in \Pi^+$ that separates data, instead of searching over $\Pi$. Intuitively, we are looking for a 'small' horosphere that captures the positive samples. We are now ready to present the Horospherical Perceptron followed by the Horospherical SVM.

## 3.3 Horospherical Perceptron

The loss function for the proposed horospherical perceptron is given by

$$l(\mu, \boldsymbol{\omega}, b; \boldsymbol{x}, y) = \max(0, -y \cdot (\mu \langle \boldsymbol{\omega}, \boldsymbol{x} \rangle_{\mathbb{B}} - b)), \quad (\mu, \boldsymbol{\omega}, b) \in \mathbb{R}^+ \times \mathbb{S}^{n-1} \times \mathbb{R}^+, \qquad (5)$$

which is zero when the instance is classified correctly and is proportional to the signed distance of the instance from the horosphere when it is misclassified. The empirical loss for a given data set $S$ is

$$L(\mu, \boldsymbol{\omega}, b) = \frac{1}{|S|} \sum_{\{\boldsymbol{x}, y\} \in S} l(\mu, \boldsymbol{\omega}, b; \boldsymbol{x}, y) \qquad (6)$$

Hence, the optimal horosphere is learned by solving the above optimization problem on the manifold $\mathbb{R}^+ \times \mathbb{S}^{n-1} \times \mathbb{R}^+$, i.e.,

$$\mu^*, \boldsymbol{\omega}^*, b^* = \arg \min_{(\mu, \boldsymbol{\omega}, b)} L(\mu, \boldsymbol{\omega}, b) \qquad (7)$$

To further analyze this optimization problem, we first recall some facts about geodesic convexity.

**Definition 3.4.** (Geodesically convex sets [30]). Let $(\mathcal{M}, g)$ be a Riemannian manifold. A set $\boldsymbol{A} \subseteq \mathcal{M}$ is said to be a geodesically convex set if, for any two points $\boldsymbol{p}, \boldsymbol{q} \in \boldsymbol{A}$, the geodesic $\gamma_{\boldsymbol{pq}}$ that connects them is contained in $\boldsymbol{A}$.

**Definition 3.5.** (Geodesically convex/concave functions [30]) Let $\boldsymbol{A} \subseteq \mathcal{M}$ be a geodesically convex set. A function $f : \boldsymbol{A} \to \mathbb{R}$ is said to be a geodesically convex function if, for any $\boldsymbol{p}, \boldsymbol{q} \in \boldsymbol{A}$ the composition $f \circ \gamma_{\boldsymbol{pq}} : [0, 1] \to \mathbb{R}$ is a convex function, where $\gamma_{\boldsymbol{pq}} : [0, 1] \to \mathcal{M}$ is a geodesic that connects $\boldsymbol{p}, \boldsymbol{q}$. $f$ is said to be a geodesically concave function if $-f$ is a geodesically convex function.

**Theorem 3.6.** *[30] Let $\boldsymbol{A} \subseteq \mathcal{M}$ be a geodesically convex set. A function $f : \boldsymbol{A} \to \mathbb{R}$ is geodesically convex if and only if its epigraph $\mathrm{epi}(f) = \{(\boldsymbol{p}, c)|f(\boldsymbol{p}) \leq c\} \subset \boldsymbol{A} \times \mathbb{R}$ is a convex set.*

Now we present the main theoretical result of this paper.

**Theorem 3.7.** *For a given training data sample $\{\boldsymbol{x}, y\} \in S, 0 < \|\boldsymbol{x}\| < R < 1$ (the hyperbolic feature $\boldsymbol{x}$ neither lie on the center nor lie on the boundary), $l(\mu, \boldsymbol{\omega}, b; \boldsymbol{x}, y)$ is a geodesically convex function on $\mathbb{R}^+ \times \boldsymbol{A} \times \mathbb{R}^+$ and is a geodesically concave function on $\mathbb{R}^+ \times \boldsymbol{B} \times \mathbb{R}^+$, where*

$$\boldsymbol{A} = \left\{\boldsymbol{\nu} \in \mathbb{S}^{n-1} \Big| y \cdot \frac{\boldsymbol{x}^T \boldsymbol{\nu}}{\|\boldsymbol{x}\|} > 0 \right\} \subset \mathbb{S}^{n-1}, \quad \boldsymbol{B} = \left\{\boldsymbol{\nu} \in \mathbb{S}^{n-1} \Big| y \cdot \frac{\boldsymbol{x}^T \boldsymbol{\nu}}{\|\boldsymbol{x}\|} < 0 \right\} \subset \mathbb{S}^{n-1}. \quad (8)$$

*Note that both $\boldsymbol{A}$ and $\boldsymbol{B}$ are geodesically convex sets.*

*Proof.* Let $l(\mu, \boldsymbol{\omega}, b; \boldsymbol{x}, y) = \max(0, -g(\mu, \boldsymbol{\omega}, b; \boldsymbol{x}, y))$ where $g(\mu, \boldsymbol{\omega}, b; \boldsymbol{x}, y) = y \cdot (\mu\langle \boldsymbol{\omega}, \boldsymbol{x}\rangle_{\mathbb{B}} - b)$. Since $\max(0, -a)$ is a convex function in $a \in \mathbb{R}$ and $g(\cdot)$ is linear in $\mu$ and $b$, we only need to show that $g(\cdot)$, as a function of $\boldsymbol{\omega} \in \mathbb{S}^{n-1}$, is geodesically convex (concave). Without loss of generality, let $\mu = 1$ and $b = 0$. Also note that $y$ is the label of data that takes values from $\{-1, +1\}$ which may flip the inequality. It suffices to validate results for the positive sample, i.e, $y = 1$.

With a slight abuse of notation, let $g(\boldsymbol{\omega}; \boldsymbol{x}) = g(1, \boldsymbol{\omega}, 0; \boldsymbol{x}, 1) = \langle \boldsymbol{\omega}, \boldsymbol{x}\rangle_{\mathbb{B}} = \ln\frac{1 - \|\boldsymbol{x}\|^2}{\|\boldsymbol{\omega} - \boldsymbol{x}\|^2}$ defined on $\boldsymbol{A} = \left\{\boldsymbol{\nu} \in \mathbb{S}^{n-1} \Big| \frac{\boldsymbol{x}^T \boldsymbol{\nu}}{\|\boldsymbol{x}\|} > 0\right\} \subset \mathbb{S}^{n-1}$. Since $-\ln(\cdot)$ is decreasing and convex, we only need to check $h(\boldsymbol{\omega}; \boldsymbol{x}) = \|\boldsymbol{\omega} - \boldsymbol{x}\|^2$ is geodesically convex on $\boldsymbol{A}$, i.e, check that the epigraph of $h$ is a convex set. Note that $\frac{\boldsymbol{x}^T \boldsymbol{\omega}}{\|\boldsymbol{x}\|} = \cos(\theta_{\boldsymbol{\omega}})$ for $\boldsymbol{\omega} \in \mathbb{S}^{n-1}$ where $\theta_{\boldsymbol{\omega}} = \angle(\boldsymbol{\omega}, \boldsymbol{x})$.

$$\begin{aligned}
\mathrm{epi}(h) &= \{(\boldsymbol{\omega}, c) \in \boldsymbol{A} \times \mathbb{R} | \|\boldsymbol{\omega} - \boldsymbol{x}\|^2 \leq c\} \\
&= \left\{(\boldsymbol{\omega}, c)\Big| \frac{\boldsymbol{x}^T \boldsymbol{\omega}}{\|\boldsymbol{x}\|} \geq \frac{1}{2\|\boldsymbol{x}\|}(1 + \|\boldsymbol{x}\|^2 - c)\right\} \\
&= \{(\boldsymbol{\omega}, c) | \cos(\theta_{\boldsymbol{\omega}}) \geq d(c)\} = \begin{cases} \boldsymbol{A} \times [d(c), \infty) & \text{if } d(c) \leq 0 \\ \boldsymbol{A}_d \times [d(c), \infty) & \text{if } d(c) > 0 \end{cases}
\end{aligned} \quad (9)$$

where $d(c)$ is a real number depending on $c$ and $\|\boldsymbol{x}\|$ and $\boldsymbol{A}_d = \{(\boldsymbol{\omega}, c) | \cos(\theta_{\boldsymbol{\omega}}) \geq d(c)\}$ is the collection of unit vectors where the angle between the vectors given the data $\boldsymbol{x}$ is small i.e., restricted to a small region on the sphere. The last equality follows from the definition of $\boldsymbol{A}$ and $\boldsymbol{A}_d$: if $d(c) \leq 0$, then $\{\boldsymbol{\omega} : \cos(\theta_{\boldsymbol{\omega}}) \geq d(c)\} \cap \boldsymbol{A} = \boldsymbol{A}$. Similarly, if $d(c) > 0$, $\{\boldsymbol{\omega} : \cos(\theta_{\boldsymbol{\omega}}) \geq d(c)\} \cap \boldsymbol{A} = \{\boldsymbol{\omega} \in \mathbb{S}^{n-1} | \cos(\theta_{\boldsymbol{\omega}}) \geq d(c)\} := \boldsymbol{A}_{d(c)}$. Both $\boldsymbol{A}$ and $\boldsymbol{A}_d$ are geodesically convex sets and this completes the proof. ∎

The convex sets $\boldsymbol{A}, \boldsymbol{B}$ are the hemispheres of $\mathbb{S}^{n-1}$ separated by the hyperplane $\{\boldsymbol{z} \in \mathbb{R}^n | \boldsymbol{x}^T \boldsymbol{z} = 0\}$ (the hyperplane has $\boldsymbol{x}$ as its normal vector) in its ambient space $\mathbb{R}^n$. Theorem 3.7 tells us that given one data sample $(\boldsymbol{x}, y)$, the optimal value described in Eq. (7) exists in $\mathbb{R}^+ \times \boldsymbol{A} \times \mathbb{R}^+$, and it is globally optimal. For a collection of training samples $S = \{(\boldsymbol{x}_i, y_i)\}_{i=1}^N$, let $\boldsymbol{A}_i = \left\{\boldsymbol{\nu} \in \mathbb{S}^{n-1} \Big| y_i \cdot \frac{\boldsymbol{x}_i^T \boldsymbol{\nu}}{\|\boldsymbol{x}_i\|} > 0\right\}$. If the data are separable by a horosphere, it follows that $\cap_{i=1}^N \boldsymbol{A}_i$ is non-empty and convex. Then the loss function given $S$ is geodesically convex on $\mathbb{R}^+ \times \cap_{i=1}^N \boldsymbol{A}_i \times \mathbb{R}^+$ and the global optimum can be obtained using any gradient-based optimization. Numerically, we apply a Riemannian gradient descent method on the entire space $\mathbb{R}^+ \times \mathbb{S}^{n-1} \times \mathbb{R}^+$ since $g(\mu, \boldsymbol{\omega}, b; \boldsymbol{x}, y)$ is continuous.

## 3.4 Horospherical SVM

Given a horospherical decision boundary $\pi_{\mu, \boldsymbol{\omega}, b}$ parameterized by $\boldsymbol{\omega} \in \mathbb{S}^{n-1}, \mu \in \mathbb{R}^+$, and $b \in \mathbb{R}$, the margin $\gamma$ is the minimal distance from training samples $S$ to the decision boundary:

$$\gamma(\mu, \boldsymbol{\omega}, b) = \inf_{\{\boldsymbol{x}, y\} \in S} y \cdot f(\boldsymbol{x}; \mu, \boldsymbol{\omega}, b) \cdot d_{\mathbb{B}}(\boldsymbol{x}, \pi_{\mu, \boldsymbol{\omega}, b}) = \inf_{\{\boldsymbol{x}, y\} \in S} y \cdot \frac{(\mu\langle \boldsymbol{\omega}, \boldsymbol{x}\rangle_{\mathbb{B}} - b)}{\mu}. \quad (10)$$

The maximum margin classifier can be obtained by solving the following optimization problem:

$$\max_{\mu, \boldsymbol{\omega}, b} \quad \gamma(\mu, \boldsymbol{\omega}, b) \qquad s.t. \quad y \cdot \frac{(\mu\langle \boldsymbol{\omega}, \boldsymbol{x}\rangle_{\mathbb{B}} - b)}{\mu} \geq \gamma \quad \text{for all } (\boldsymbol{x}, y) \in S. \quad (11)$$

**Theorem 3.8.** *The maximum margin classification problem in hyperbolic space with horosphere as its decision boundary described in Eq.* (11) *is equivalent to the following optimization problem:*

$$\min_{\mu, \boldsymbol{\omega}, b} \quad \frac{1}{2}\mu^2 \qquad s.t. \quad y \cdot (\mu \langle \boldsymbol{\omega}, \boldsymbol{x} \rangle_{\mathbb{B}} - b) \geq 1 \quad \text{for all } (\boldsymbol{x}, y) \in S. \tag{12}$$

The proof is analogous to that in the Euclidean case [3]. Note that the margin is unchanged if we apply the following scale transformation: $\mu \to \mu/\gamma$ and $b \to b/\gamma$. We can also build a soft-margin horospherical SVM, dubbed HoroSVM, by minimizing the following loss function:

$$l(\mu, \boldsymbol{\omega}, b; \boldsymbol{x}, y) = \frac{1}{2}\mu^2 + C \sum_{i=1}^{|S|} \max(0, 1 - y_i \cdot (\mu \langle \boldsymbol{\omega}, \boldsymbol{x}_i \rangle_{\mathbb{B}} - b)). \tag{13}$$

where $C$ is a hyperparameter that controls the tradeoff between minimizing misclassification and maximizing margin.

It is easy to see that the same result as in Theorem 3.7 holds for the loss function in Eq. (13). That is, the loss function is a geodesically convex function on one geodesically convex subset of the parameter space and a geodesically concave function on the other geodesically convex subset of the parameter space. Recall that the idea behind proving that the loss function in the horospherical perceptron, $\max(0, -g(\cdot))$, where $g(\cdot) = y \cdot (\mu \langle \boldsymbol{\omega}, \boldsymbol{x} \rangle_{\mathbb{B}} - b)$, is geodesically convex is based on the important fact that $\max(0, -a)$ is a convex function in $a \in \mathbb{R}$. Similarly, the same idea applies to HoroSVM, where the hinge loss is used, and the loss function becomes $\frac{1}{2}\mu^2 + \max(0, 1 - g(\cdot))$. Note that $\max(0, 1 - a)$ is also a convex function in $a \in \mathbb{R}$ and $\frac{1}{2}\mu^2$ is a convex function in $\mu$, these facts complete the proof of the desired property for the HoroSVM, which is analogous to Theorem 3.7 for horospherical perceptron.

We can then apply any Riemannian gradient descent optimization methods for updating the parameters in HoroSVM since the problem is an optimization problem over a product space of Riemannian manifolds, $\mathbb{R}^+ \times \mathbb{S}^{n-1} \times \mathbb{R}^+$. We refer the readers to [5] and [1] for more details about optimization techniques on Riemannian manifolds.

## 4 Experiments

In this section, we present several experimental results obtained from an application of our HoroSVM to synthetic data as well as real data sets used in published literature. Our implementation is based on Pymanopt [19] using the Riemannian conjugate gradient method [26] on Intel(R) Xeon(R) CPU E5-2683 v3 @ 2.00GHz.

### 4.1 Network Data Set

Here, we follow the experimental setup in [12], and evaluate our HoroSVM over four real-world network data sets used by [8]: `karate` [34] (2 classes, 34 nodes ), `polblogs` [2] (2 classes, 1224 nodes ), `polbooks` [3] (3 classes, 105 nodes ), and `football` [17] (12 classes, 115 nodes ).

The network data is embedded in a 2D hyperbolic space using the method of [8]. Given the hyperbolic embeddings, we compare our HoroSVM with three other competing large margin classifiers: Euclidean SVM (even though it violates the hyperbolic geometry), hyperboloid SVM [12], and Poincaré SVM [11]. The absence of comparison with [32] in this experiment is due to two reasons. First, [32] aims to provide a theoretical understanding of hyperbolic spaces in classification, focusing on 'linearly' separable data (data that can be separated by a geodesic). They do not address extensions to nonlinearly separable data, which limits their applicability to many practical datasets. Second, for datasets that are linearly separable in hyperbolic space, the approach in [32] aligns with the hyperboloid SVM when adversarial training is not applied in [32].

For multiclass classification, a one-verses-rest strategy is applied. We conducted a five-fold cross-validation on each data set, where we chose the hyperparameter $C$ from $\{1, 5, 10\}$ during the cross-validation procedure. Note that a more extensive search space for $C$ may lead to potential performance improvements. The mean of the F1 score followed by the standard deviation over five

---

[3] http://www-personal.umich.edu/~mejn/netdata/

Table 1: F1 scores for node classification on network datasets. Boldface indicates best performance.

| Methods | Karate | Polblogs | Polbooks | Football |
|---|---|---|---|---|
| Euclidean SVM | $0.95 \pm 0.06$ | $0.92 \pm 0.02$ | $0.83 \pm 0.03$ | $0.29 \pm 0.12$ |
| Hyperboloid SVM | $0.95 \pm 0.06$ | $0.92 \pm 0.01$ | $0.83 \pm 0.03$ | $0.30 \pm 0.14$ |
| Poincaré SVM | $0.78 \pm 0.16$ | $0.92 \pm 0.02$ | $0.84 \pm 0.03$ | $0.32 \pm 0.04$ |
| HoroSVM (Ours) | $\mathbf{0.98 \pm 0.04}$ | $\mathbf{0.93 \pm 0.01}$ | $\mathbf{0.85 \pm 0.04}$ | $\mathbf{0.34 \pm 0.06}$ |

Table 2: F1 scores for subtree classification on four subtrees of WordNet. Boldface indicates the best performance on 2D embeddings of each dataset.

| Methods | animal.n.01 3218/798 | group.n.01 6649/1727 | worker.n.01 861/254 | mammal.n.01 953/228 |
|---|---|---|---|---|
| Hyperboloid SVM (D = 2) | $0.53 \pm 0.07$ | $0.52 \pm 0.01$ | $0.54 \pm 0.04$ | $0.39 \pm 0.03$ |
| Hyperbolic LR (D = 2) | $0.46 \pm 0.08$ | $0.52 \pm 0.04$ | $0.54 \pm 0.07$ | $0.32 \pm 0.10$ |
| Hyperbolic LR (D = 5) | $0.95 \pm 0.03$ | $0.76 \pm 0.07$ | $0.80 \pm 0.08$ | $0.78 \pm 0.04$ |
| Hyperbolic LR (D = 10) | $0.96 \pm 0.01$ | $0.86 \pm 0.05$ | $0.84 \pm 0.04$ | $0.94 \pm 0.04$ |
| Euclidean SVM (D = 2) | $0.39 \pm 0.01$ | $0.39 \pm 0.00$ | $0.32 \pm 0.02$ | $0.20 \pm 0.01$ |
| Euclidean SVM (D = 5) | $0.95 \pm 0.00$ | $0.79 \pm 0.01$ | $0.38 \pm 0.02$ | $0.44 \pm 0.01$ |
| Euclidean SVM (D = 10) | $0.97 \pm 0.00$ | $0.91 \pm 0.00$ | $0.46 \pm 0.04$ | $0.72 \pm 0.05$ |
| HoroSVM (D = 2) | $\mathbf{0.57 \pm 0.07}$ | $\mathbf{0.65 \pm 0.01}$ | $\mathbf{0.62 \pm 0.01}$ | $\mathbf{0.42 \pm 0.01}$ |
| HoroSVM (D = 5) | $0.93 \pm 0.01$ | $0.88 \pm 0.00$ | $0.82 \pm 0.04$ | $0.88 \pm 0.01$ |
| HoroSVM (D = 10) | $0.95 \pm 0.02$ | $0.91 \pm 0.01$ | $0.86 \pm 0.01$ | $0.93 \pm 0.02$ |

trials are summarized in Table 1. As evident from the table, our method yields the best results on all the data sets.

HoroSVM outperformed other methods on all four data sets. The data in `karate` are well-separated and thus both Euclidean SVM and hyperboloid SVM performed equally well. Our method outperforms the others since the horospheres have several nice properties, the most important of which is that the Busemann function whose level sets are the horospheres is a convex function that guarantees global optimality in the optimization. Notice that the performance of Poincaré SVM on `karate` is inferior to others by a significant amount. The reason is that the performance of Poincaré SVM is sensitive to the choice of the reference point. We demonstrate our performance gain on the remaining data sets, and our method is more consistent, compared to Euclidean SVM and hyperboloid SVM, in terms of lower standard deviation, on `football` data set where data exhibit a larger variance/spread.

## 4.2 Subtree Classification in WordNet

A task of considerable interest in hyperbolic space classification problems is to determine whether a node belongs to a given subtree in the hyperbolic embedding. We obtained hyperbolic embeddings in various dimensions using the approach in [15] for WordNet [4] noun hierarchy (82,115 nodes). We consider four subtrees whose roots are the following synsets: ANIMAL.N.01, GROUP.N.01, WORKER.N.01, and MAMMAL.N.01.

We split all nodes in a subtree into positive training (80%) and test (20%) nodes and applied the same process to the remaining WordNet nodes to create negative training and test sets. The average F1 scores and the standard deviations over 3 trials are shown in Table 2. The number of positive training/test samples of each data set are listed as well. We exclude Poincaré SVM from the comparisons in this task. The reason being, data are highly imbalanced in this task and the positive samples are clustered near the boundary. The reference point learned in Poincaré SVM will be close to the boundary where the tangent approximation of data at this reference point is highly distorted, as opposed to the original hyperbolic embeddings. It is therefore hard to locate a hyperplane in the

---

[4] https://wordnet.princeton.edu/

tangent space that separates the lifted (mapped) data. In addition, the learning of the reference point is only applicable to 2D hyperbolic space.

As well known in the Euclidean SVM literature, a vanilla (unweighted) implementation of SVM performs poorly on extremely imbalanced data, we observed the same behavior in training HoroSVM on this task. We preprocessed the data by downsampling the majority class of samples (the negative samples) to train a robust model. Note that our HoroSVM can be naturally extended to a class/instance-weighted version by assigning a class/instance weight to the penalty term $C$ for each sample, allowing us to address more general imbalanced data. Since there is no protocol for dealing with unbalanced data in training a hyperboloid SVM, we presented the Euclidean SVM results using the same preprocessed data for reference. In addition, we presented results of hyperbolic logistic regression (LR) [15], which is not a large-margin classifier on this task, where the imbalanced data is handled by sampling the equal number of negative and positive nodes in each mini-batch of size 16 during training.

Now, we highlight several results in Table 2. The superior performance of hyperboloid SVM over hyperbolic LR is expected, as both methods use geodesic decision boundaries but hyperboloid SVM aims to maximize the margin. However, the training of hyperboloid SVM is highly unstable as we mentioned earlier due to the non-convex optimization process. Euclidean SVM under-performs as it does not take into account the hyperbolic geometry. Our HoroSVM exhibits a significant improvement in predicting words in a subtree, as evidenced by higher F1 scores across all the subtrees. The small number of nodes within a subtree, compared to the whole WordNet, causes the nodes to cluster near the boundary in their hyperbolic embeddings. Thus, horosphere is an ideally suited decision boundary (in comparison to the geodesic boundary in [12]) to isolate the subtree.

### 4.3 Synthetic Data with Noisy Labels

To demonstrate the robustness of our HoroSVM, we apply it to synthetic data with noisy labels at varying levels/amounts of noise. Specifically, we generated 100 synthetic datasets by sampling from a Gaussian mixture model defined on the Poincaré disk model as in [12]. The isotropic Gaussian distribution in hyperbolic space is referred to as the *Riemannian normal* distribution, and we used the sampling method presented in [21]. For each dataset, we

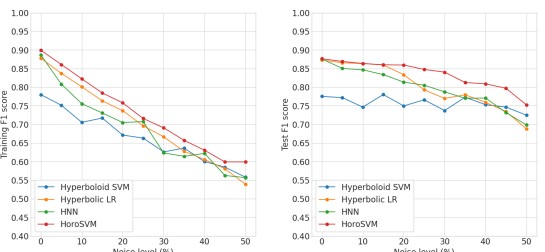

Figure 4: Training (left) and test (right) F1 scores of several methods on synthetic data with noisy labels at different noise levels.

sampled two centroids from a zero-mean Riemannian normal distribution with a variance of 1.5. We then sampled 200 data points from a unit-variate Riemannian normal distribution centered at each centroid, resulting in a dataset of 400 points classified into positive and negative classes. We split the dataset into training and test sets with 100 positive/negative samples in the training set and 100 positive/negative samples in the test set. We then generated datasets with noisy labels at noise levels: $\eta \in \{0, 0.05, 0.1, \ldots, 0.5\}$ by flipping the labels of a proportion $\eta$ of the training (not test) samples, with an equal number of positive and negative samples flipped. The train/test average F1 scores of each method across all datasets at varying noise levels are shown in Fig 4. We compared our HoroSVM with hyperboloid SVM, hyperbolic LR, and a two-layer HNN [15] (with a hidden dimension of 5). While all methods depict decreasing training F1 scores as the noise level increases, HoroSVM outperforms the others consistently throughout the training process. Hyperbolic LR and HNN exhibit the least resistance to label noise, with test F1 scores dropping (faster) with increasing noise level. Both hyperboloid SVM and HoroSVM demonstrate consistent performance across different noise levels, owing to the inherent robustness of large-margin classifiers. However, the training of hyperboloid SVM is highly unstable resulting in its inferior performance. HoroSVM demonstrates its superiority in accuracy and robustness as evidenced in the results. In addition, we present the average training times for each method on one dataset (200 samples) as follows: 6.57 seconds (Hyperboloid SVM), 3.98 seconds (Hyperbolic LR), 9.06 seconds (HNN), and 3.73 seconds (HoroSVM). Notably, HoroSVM stands out as the fastest.

# 5 Discussion and Conclusions

In this paper, we presented a novel large margin classifier, dubbed HoroSVM, whose decision boundaries are horospheres that are the level sets of a Busemann function. We presented a novel formulation leading to the optimization of a geodesically convex loss performed using a Riemannian gradient-based method and guaranteeing a globally optimal solution. We demonstrated superior to competitive performance of the HoroSVM over SOTA large margin classifiers.

In Euclidean space, a kernel SVM is usually favored over the linear SVM due to its ability to cope with non-linearly separable data. In Hyperbolic space, the challenge lies in developing valid positive definite kernels (see [14] for details on validity of kernels on Riemannian manifolds). The only reported work on KSVM in hyperbolic space that we are aware of is [12], which uses a kernel that violates the positive definiteness property of RKHS kernels. Thus the problem of interest is primarily defining a valid family of kernels in hyperbolic space. We will address this in our future work.

## Acknowledgements

This research was in part funded by the NSF grant IIS 1724174 and the NIH NINDS and NIA grant RF1NS121099 to Vemuri.

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
