# OpenReview forum: "Horospherical Decision Boundaries for Large Margin Classification in Hyperbolic Space"
_NeurIPS.cc/2023/Conference — NeurIPS 2023 poster_

### Official Review · Reviewer_38iz · 2023-07-06

**Soundness:** 3 good
**Presentation:** 3 good
**Contribution:** 3 good
**Rating:** 7
**Confidence:** 4

**Summary:**

This work studies the support vector machines for data with latent hierarchical relationships, which are represented in hyperbolic spaces. Similar to the linear SVM in Euclidean spaces, the SVM in hyperbolic spaces have been seen in the literature but they are challenging due to some issues such as non-convexity or algorithmic convergence. In this work, the authors explored the analogs between horospheres and Euclidean hyperplanes, and proposed to use geodesic segment (counterpart of the Euclidean distances) to define the margin for the SVM. A nice theory shows the resulting problem is convex so a stable algorithm is developed. Numerical studies demonstrate the superior performance over some competitors.

**Strengths:**

This paper is working on an integral question for classifying network data. It introduces an interesting idea of defining the SVM on horospheres and shows good performance on benchmark data. In general, the paper appears to be technically correct with sound theory for showing convexity.

**Weaknesses:**

1. The authors may comment on the computational speed of the proposed algorithm, as it may outperform Hyperboloid SVM due to its convexity.
2. From Table 2, it seems HoroSVM well outperforms the other classifiers when D = 2, and the advantage gets smaller when D is large. What is the performance of HoroSVM when D is larger?  Furthermore, it would be useful to discuss how D is chosen in real-world scenarios.
3. The authors should also compare with the kernel SVM to see if the latent data structure can be implicitly handled by the kernel SVM.
4. Most datasets chosen for the study display a significant imbalance. It would be interesting to see if it is possible to incorporate cost-sensitive weights into these observations. Doing so could mirror the approach taken by weighted SVM to tackle the imbalance issue.



**Questions:**

In Section 4.1, the hyperparameter C is selected from {1, 5, 10}. It might be helpful to comment on how sensitive this parameter is. The performance of a linear SVM is typically highly sensitive to this parameter, and the paper could potentially mislead readers by implying that a non-rigorous selection from only three candidate values would suffice for optimal performance.

---

> ### Author Rebuttal · Authors · 2023-08-09
>
> # Response to Reviewer 38iz
>
> We thank the reviewer for the valuable comments and questions. Please find our responses below:
>
> > 1. computational speed of the proposed algorithm
>
> The complexity of our approach depends on the Riemannian gradient-based optimization technique that we chose to use. In [36], a comprehensive analysis of the complexity of first-order methods was presented. In our case, we use the Riemannian conjugate gradient method, whose algorithm and global convergence analyses are presented in [C2]. Hyperboloid SVM, where a non-convex optimization problem is solved using projected gradient descent, results in a potential local optimum.
>
> Furthermore, we will include the running time on our machine (specs. given in the paper) for each method in Experiment 3 as a reference to the time complexity in the revision. The average training time on our machine for each method on a dataset with 200 samples is as follows: 6.57s (hyperboloid SVM), 3.98s (hyperbolic LR), 9.06s (HNN), and 3.73s (HoroSVM). Notably, our HoroSVM stands out as the fastest.
>
>
> > 2. It seems HoroSVM well outperforms the other classifiers
> when D = 2, and the advantage gets smaller when D is large
>
> It is well known that entangled data in low dimensions may be easier to separate by raising the dimensions. As evident in the  experiments, some of the competing methods yielded higher accuracy improvements with increasing dimensions compared to our HoroSVM. This is to be expected since HoroSVM performed the best in terms of classification accuracy for $D=2$ and hence there was not much room for further accuracy gain to be achieved by increasing dimensions in the HoroSVM case. The strength of the HoroSVM lies in the ability to achieve higher accuracy classification even in lower dimensions where data might be highly entangled. We focus on low dimensions since, as the dimension increases (from 10 to 200), the performance gain of hyperbolic embeddings of graphs is marginal, as empirically evaluated in [23]. Our superior performance in low dimensions is by virtue of the property of hyperbolic space being efficient in representing hierarchical data. Theoretical analysis of our method's generalization error as a function of dimension is an interesting problem and will be the focus of our future work.
>
> > 3. compare with the kernel SVM
>
> We would like to point out that the only reason for our comparison with linear SVM in Euclidean space is primarily to show that ignoring the geometric structure of data space leads to poorer performance. We will include this reason in the revision. To perform a fair comparison with kernel SVM, one would have to develop a kernel version of SVM in hyperbolic space. We will explore this idea in future work.
>
> > 4. Most datasets chosen for the study display a significant imbalance.
>
> We have indeed encountered sensitivity while training our HoroSVM on highly imbalanced datasets, similar to the challenges faced by Euclidean SVM. In this work, we used a downsampling strategy that will remove the imbalance at the cost of losing some data but, since the WordNet dataset contains a sufficient number of samples in the minority class, this doesn't significantly affect the classifier  performance.
>
> To deal with more general imbalanced data, our HoroSVM can be naturally extended to a class/instance-weighted version by assigning a class/instance weight to the penalty term $C$ for each sample, just as it is done in Euclidean SVM.
>
> The idea of extending our work to a cost-sensitive SVM, as explored in [C3] for Euclidean space, holds significant appeal and promise. We will investigate this direction in future research.
>
> > 5. the hyperparameter C is selected from 1, 5, 10. It might be helpful to comment on how sensitive this parameter is.
>
> The choice of hyperparameter $C$ is crucial in HoroSVM as it is in linear SVM since it balances misclassification and margin maximization. As is standard in Euclidean SVM, we employ grid search to select the hyperparameter $C$ from a set of candidate values in our method. We acknowledge that this limited set of candidates might not fully explore the optimal range for $C$, and we will include this discussion in the revision.

---

### Official Review · Reviewer_1ZfQ · 2023-07-07

**Soundness:** 2 fair
**Presentation:** 3 good
**Contribution:** 2 fair
**Rating:** 4
**Confidence:** 4

**Summary:**

This paper presented a novel large margin classifier, dubbed HoroSVM, whose decision boundaries are horospheres in hyperbolic space, and proved it’s a convex optimization problem.
This paper presented several experiments depicting the competitive performance of the classifier in comparison to SOTA.


**Strengths:**

1. This paper presented a novel large margin classifier, dubbed HoroSVM, whose decision boundaries are horospheres in hyperbolic space. It’s innovative compared to the predecessors.
2. This paper systematically and clearly explain the proof, which is easy to follow.


**Weaknesses:**

1. The experiments “Synthetic Data with noisy Labels” are over synthetic data, which is not convincing enough compared to apply random perturbations over real-world dataset
2. The motivation of classification in hyperbolic space is explained inexplicit.


**Questions:**

The method in this paper and the compared methods are basically based on SVM. Wouldn’t the result better if neural networks are applied to classify in hyperbolic space?

**Limitations:**

Yes

---

> ### Author Rebuttal · Authors · 2023-08-09
>
> # Response to Reviewer 1ZfQ
>
> We thank the reviewer for the valuable comments and questions. Please find our responses below:
>
> > 1. random perturbations over real-world dataset
>
> The noisy label experiment was conducted using synthetic data, which
> allows us to have full control over the level of noise injected in the labels. For the
> real word dataset, we conducted additional noisy label experiments on two
> binary-class network datasets used in Experiment 1 (karate and polblogs)
> with a 20% noise level to demonstrate the robustness of our classifier. Our
> method performs the best as evidenced in the following table.
>
> | Dataset         | Karate | Karate-Corrupted | Polblogs | Corrupted-Polblogs |
> |-----------------|--------|------------------|----------|--------------------|
> | Euclidean SVM   | 0.95   | 0.90             | 0.92     | 0.91               |
> | Hyperboloid SVM | 0.95   | 0.76             | 0.92     | 0.87               |
> | Poincare SVM    | 0.78   | 0.70             | 0.92     | 0.77               |
> | HoroSVM         | 0.98   | 0.91             | 0.93     | 0.92               |
>
>
> > 2. The motivation of classification in hyperbolic space is explained inexplicit.
>
> Hyperbolic spaces offer compact representation for hierarchical data, which provides a provable advantage over Euclidean spaces in terms of lower distortion. Classification naturally emerges as one of the standard downstream tasks for data represented in hyperbolic space A naive application of Euclidean methods to hyperbolic data (by regarding the data points to be in Cartesian coordinates) is inadequate, as it neglects the intrinsic geometry of the data. We then aim to design a classification algorithm in the hyperbolic space that respects the underlying hyperbolic geometry and achieves superior performance.
>
>
> > 3. The method in this paper and the compared methods are basically based on SVM.
>
> Hyperbolic neural networks may yield higher classification accuracy by providing a holistic solution for classification tasks, encompassing the extraction of rich hyperbolic features from complex network structures and the subsequent application of hyperbolic logistic regression (LR) [16] for final predictions. Our primary focus within this study is not on competing at the feature extraction level – that is, not evaluating which method extracts superior features – but rather on identifying the method that excels as a 'linear' large-margin classifier in hyperbolic space. Moreover, HNNs are not large-margin classifiers and hence do not possess good generalization abilities. Large margin classifiers have a well-established theory and provide performance guarantees. We have provided a framework for such a theory in the hyperbolic space by providing a formulation for the optimization that is geodesically convex and is guaranteed a globally optimal solution similar to the Euclidean SVMs. Thus our comparison is specifically focused on the class of SVMs.

---

> > ### Comment · Reviewer_1ZfQ · 2023-08-21
> >
> > Thanks for the response. From the results in the table, there seems no distinct performance gap between HoroSVM and Euclidean SVM, and the paper should also compare the HoroSVM with Hyperbolic neural networks.

---

> > > ### Author Response · Authors · 2023-08-21
> > >
> > > # Response to Reviewer 1ZfQ #2
> > >
> > > We thank the reviewer for the comment. Please find our further clarification below.
> > >
> > > We acknowledge the lack of a significant gap between Euclidean SVM and HoroSVM in this noisy label real data example. The reason is, some network data (Polblogs and Karate in Table 1) after embedding in the hyperbolic space are well separated, and using a Euclidean SVM on the ambient Euclidean coordinates of these embedded data points yields reasonably high accuracy in classification, which is also observed in hyperboloid SVM [12]. However, note that we already showed in this submission that for other real data experiments (see Tables 1&2), Euclidean SVM performance is poor compared to HoroSVM.
> > >
> > > As for the comparison to HNNs, we would like to emphasize that even in the published HNN work [16], classification of data embedded in hyperbolic space was achieved using hyperbolic logistic regression [16, Table 2 & Sec 4 Paragraph "MLR (multiclass logistic regression) classification experiments."] and not in an end-to-end fashion, i.e., learn features and classify. Since this is the SOTA in HNN, we compared our work to hyperbolic LR [16] in the WordNet real data classification (Table 2).
> > >
> > > Despite the above reasoning, upon your insistence, we used a simple end-to-end HNN model (same as the one used in producing the results in Figure 4) and tested it on noisy label real data presented earlier in this rebuttal. The results are included in the last row of the table below. As evident from the last row, HNN performs poorly on the karate dataset as well as in the Corrupted-Polblogs. We believe that comparing HoroSVM to HNN will in general involve developing a novel architecture distinct from the aforementioned simple HNN architecture and this task is outside the scope of our work.
> > >
> > >
> > > | Dataset         | Karate | Karate-Corrupted | Polblogs | Corrupted-Polblogs |
> > > |-----------------|--------|------------------|----------|--------------------|
> > > | Euclidean SVM   | 0.95   | 0.90             | 0.92     | 0.91               |
> > > | Hyperboloid SVM | 0.95   | 0.76             | 0.92     | 0.87               |
> > > | Poincare SVM    | 0.78   | 0.70             | 0.92     | 0.77               |
> > > | HoroSVM         | 0.98   | 0.91             | 0.93     | 0.92               |
> > > | HNN             | 0.91   | 0.80             | 0.92     | 0.87               |
> > >
> > >
> > > Furthermore, we have made comparisons to HNN in Figure 4 on synthetic noisy data, where noise levels can be controlled. We demonstrated the superior performance of HoroSVM over HNN, hyperbolic LR, and hyperboloid SVM.
> > >
> > > ***
> > > Reference:
> > >
> > > [12] Cho et al. Large-margin classification in hyperbolic space. AISTATS 2019.
> > >
> > > [16] Ganea et al. Hyperbolic neural networks. NeurIPS 2018.

---

### Official Review · Reviewer_yt6v · 2023-07-09

**Soundness:** 2 fair
**Presentation:** 3 good
**Contribution:** 3 good
**Rating:** 5
**Confidence:** 3

**Summary:**

The paper proposes a large margin classifier in hyperbolic space, Poincare ball models. To this end, horospherical decision boundaries (which are based on the Buseman function and are different level sets of the Busmann function at the ideal point) are used for the large-margin classifier. They also show that the formed classifier is convex and optimization can be performed using Riemannian gradient descent. They perform experiments on four datasets and compare the results with Euclidean Hyperboloid, Poincare, and Horo SVM.

**Strengths:**

The paper proposes a large margin classifier based on the horospheres. The research question is valid and the paper is well motivated, and well written. The paper also provides the proofs for the theoretical claims as well. Figures are also beneficial to understand, for example figure 2 is beneficial to compare the horocycle vs geodesic decision boundary.

The experiments also show the efficacy of the proposed approach specially on 2dimensional embedding space. Synthetic data with noisy labels is also an interesting experiment showing the robustness of the proposed approach and generally hyperbolic version to the noisy data.

The analysis on the performance on the experiments section is beneficial as well.


**Weaknesses:**

There is one main question about comparison with [33]. The paper generally is well written , there a few questions which I will add to the questions section.

I would suggest the authors to do a proofreading. The paper discusses horosphere decision boundary in several different ways like horospheres decision boundary, horocycle decision boundary, and horospherical decision boundary. Using a fixed terminology throughout the paper can increase the comprehensiveness of the paper.

**Questions:**

Why does not the paper compare the method with [33] in the experiments?
Lines 176- 184 are not very clear. How are the positive and negative $\pi$ subsets of the same one? shouldn't they be on the opposite sides of the Poincare ball model? Why do the authors claim that the positive samples are clustered near the boundaries?

What is the reference point in Poincare SVM? and why is it sensitive to the reference point?

**Limitations:**

The paper discusses future work shortly in the conclusions.

---

> ### Author Rebuttal · Authors · 2023-08-09
>
> # Response to Reviewer yt6v
>
> We thank the reviewer for the valuable comments and questions. Please find our responses below:
>
> > 1. comparison with [33]
>
> The reason for the absence of comparison with Weber et al. [33] in our experiments are:
>
>
> (1). Weber et al. [33, Sec 6]  focus on theoretically understanding the benefits of hyperbolic spaces for classification, using experiments to illustrate the results derived for linear models and separable data. Extensions to non-separable data and non-linear models are not addressed in [33].
>
> (2). For datasets that are separable in hyperbolic space, the approach in Weber et al. [33, Sec 6 & Fig 3] is aligned with the approach of hyperboloid SVM [12] under specific constraints namely, when the adversarial budget $\alpha =  0$. Further elaboration on this point is provided below.
>
> In [33], authors employ the same parameterization of the hyperbolic geodesic decision plane for classification in hyperbolic space as utilized in hyperboloid SVM [12]. It is shown theoretically in [33] that a hyperbolic perceptron employing such geodesic decision boundaries is guaranteed to converge on separable data using a gradient descent method. Subsequently, [33, Sec 3.2] demonstrates that by employing margin losses, such as the logistic or hinge loss, in the optimization process, a large-margin classifier can be effectively learned through gradient descent. It is noteworthy that this corresponds to hyperboloid SVM [12] when the hinge loss is applied.
>
> Moreover, [33] establishes a method to inject adversarial examples into the gradient-based loss minimization process, leading to an algorithm that efficiently learns a large-margin classifier. In this approach, the magnitude of the perturbation added to the original example is bounded by an adversarial budget $\alpha$. When $\alpha=0$, the method in [33] aligns with the one presented in [12].
>
> When dealing with real-world datasets where the assumption of data separability is scarcely fulfilled, the applicability of [33] becomes limited. For experiments on real data where the assumption that the data is separable is barely fulfilled, [33] is not applicable and we compare our method with [12]. Additionally, we will explore the adoption of an adversarial setting for our method in future work.
>
> > 2. proofreading
>
> We highly appreciate your comments and will align the terminology accordingly in the revision.
>
> > 3. Lines 176- 184 are not very clear.
>
> In Euclidean space, if $n$ is the normal vector of a hyperplane, $-n$ is also the normal vector of this hyperplane. In hyperbolic space, however, let $\Pi_{\omega}$ be the set of horospheres tangent at $\omega$, then $\Pi_{-\omega}$ is the set of horospheres tangent at $-\omega$, the antipodes of $\omega$ on $\partial\mathbb{B}^n$. Note that $\Pi_{\omega} \neq \Pi_{-\omega}$. For a given $\omega$, $\Pi^+$ is the collection of horospheres that are tangent at $\omega$ and do not contain the origin (spheres with radius $<$ 0.5), while $\Pi^-$ is the collection of horospheres that are tangent at $\omega$ and contain the origin (spheres with radius $>$ 0.5).
>
> As the volume grows exponentially in hyperbolic space, there is more space/volume as we approach the boundary and hence more data can be accommodated near the boundary. Hence, for a tree embedded in hyperbolic space, the leaf nodes within a subtree appear as a cluster near the boundary, and the clusters (representing different subtrees) are well separated from each other. Hence in the subtree classification task, we can properly assign the positive labels to a subtree of interest.
>
> > 4. reference point in Poincare SVM
>
> In Poincare SVM [11], the reference point $p$ serves as the anchor point to which all data are lifted into the tangent space at $p$. The classification is performed in that tangent space. However, the data distortion varies across different tangent spaces, making the performance of Poincare SVM sensitive to the choice of the reference point.

---

> > ### Comment · Reviewer_yt6v · 2023-08-20
> > **Answer to rebuttal**
> >
> > Thank you for providing the answers,  I will keep my current score. I would recommend the authors to include the explanation of differences with [33] in the paper as well.

---

### Official Review · Reviewer_cEx8 · 2023-07-13

**Soundness:** 2 fair
**Presentation:** 2 fair
**Contribution:** 3 good
**Rating:** 6
**Confidence:** 3

**Summary:**

Based on recent successes with hyperbolic embeddings of data with a (latent) hierarchical structure, this paper proposes a new type of SVM in hyperbolic space. Their SVM, named HoroSVM, uses horospheres as decision boundaries and the authors derive a way to compute the distance of a point to such a horosphere. This makes the method different from other recently proposed hyperbolic SVMs, which use geodesic hyperplanes as decision boundaries. The authors argue that these horospheres are a more suitable generalization of hyperplanes than their geodesic-based counterparts and that, therefore, HoroSVM is likely to outperform the other methods.

After introducing their method, the authors show that their loss function is geodesically convex with respect to the learnable parameters. As a result, they state that a global optimum to the optimization problem, that arises when using HoroSVM, can be found using gradient-based optimization methods. They further state that, while their method has this nice property, the other methods to which they compare do not. Therefore, HoroSVM should again yield better results.

With a series of three experiments, the authors attempt to empirically show the superiority of their method over geodesic hyperplane methods and Euclidean SVM. The first experiment aims to show the superior performance on the classification of hyperbolic embeddings of network data. The second experiment shows that HoroSVM is better than its competitors at predicting the subtree to which nodes belong within WordNet. The third and final experiment shows that their method is more robust than their competitors to noisy labels.

Lastly, based on their theoretical analysis and empirical results, the authors conclude that their method is more performant and robust than its competitors, while also alluding to future work with hyperbolic kernel SVMs.



**Strengths:**

The paper contains several, mostly theoretical strengths in my opinion:

1. The paper derives a way to compute distances from points on the Poincaré ball to horospheres. While directly relevant to their HoroSVM method, this formulation is likely useful to any other method that implements these horospheres on the Poincaré ball, making it a nice contribution.
2. The authors prove the geodesic convexity of their newly proposed loss function, resulting in a very strong statement about the possibility of finding a global optimum to their optimization problem. If it is indeed the case that the competitive hyperbolic methods do not have this property, then this is a very clear theoretical argument for the superiority of their method w.r.t. these other hyperbolic methods.
3. In my opinion the paper introduces an interesting theoretical analysis of their method, which could by itself be useful as a tool for studying other methods in hyperbolic space or even on other Riemannian manifolds.
4. The experiments show the superiority of their method with respect to their competitors both w.r.t. classification performance and noisy label robustness.

**Weaknesses:**

There are a few concerns that I have with this paper, which mostly have to do with the motivation for using horocycles over geodesic hyperplanes and with the setup of the experiments. Firstly, my concerns with the motivation for the use of horocycles are:

1. There are multiple geometric definitions of hyperplanes possible and it is not clear which one is chosen here. However, it is stated that horocycles are the hyperbolic equivalent of Euclidean hyperplanes and that, therefore, horocycles are a natural choice for decision boundaries. This does not provide a clear motivation for choosing horocycles over geodesic hyperplanes. What is the geometric motivation for this choice and why should it lead to better results in practice?
2. Figure 2 is supposed to provide a clear motivation for why horocycles are better as decision planes than geodesic hyperplanes, but the geodesic hyperplane example (on the right) seems poorly chosen to me. In fact, it seems very easy to find a hyperplane by hand that would lead to significantly better results than the hyperplane in the figure. What is going on here? Is this really a problem with geodesic hyperplanes or is it a problem with the optimization process. If the latter is the case, then this is still not a direct argument for using horocycles over geodesic hyperplanes.


My concerns regarding the experiments (mostly experiment 1) are:

1. In the experiment from Subsection 4.1, it is unclear how the hyperbolic embeddings of the network data were generated. As the method for generating these embeddings is likely to affect the outcome of the experiment, it is difficult to judge its validity without a description hereof.
2. Moreover, after the 5-fold cross-validation, the differences between the results obtained with the different methods are quite small for at least two of the datasets, especially compared to the standard deviation. This makes the experiment seem a bit weak as a motivation for using HoroSVM.
3. Also, given that solving the optimization problem for HoroSVM leads to a global optimum, why do you not simply choose the optimal C? And in that case, would the standard deviation not simply be 0?
4. What happens when you use a Euclidean embedding of the data and then Euclidean SVM? Even though the Euclidean embeddings would be distorted, it seems plausible to me that Euclidean SVM could obtain better results in such a setting. The current comparison does not seem completely fair to me.
5. In the second experiment the embeddings are obtained through the application of hyperbolic entailment cones. However, there are some issues with this method and there is a method from the paper "Representation Tradeoffs for Hyperbolic Embeddings" by de Sa et al. which boasts greater performance than hyperbolic entailment cones. How do the results change if this method had been used to generate the embeddings?
6. Lastly, in Table 2 the focus is put on the case of low dimension D = 2, where HoroSVM is significantly better. However, for this low dimension the performance of every method is rather low and for higher dimensions the difference in performance seems to fade. Is there a reason to still focus on low dimensions? What happens if the dimension is chosen even greater? Does the advantage of HoroSVM disappear completely?


**Questions:**

Alongside the questions posed in the weaknesses above, I have a few additional questions:

1. Why are there both a mu and a b parameter in equation (2)? You only need 1 parameter here. Is it for notational simplicity later on? If so, a small note might help to clarify this for the reader. If not, removing this over-parameterization would also make the different versions of pi with subscripts a bit easier to follow.
2. Theorem 3.7 tells us that, for a single sample, the loss function is geodesically convex w.r.t. the learnable parameters. Is this result strong enough to guarantee convergence to a global optimum in case of a collection of samples? Based on the discussion in the introduction I presume it is, but I think it would be nice to mention this for the reader.
3. Why is the Poincaré inner product named an inner product? This name seems confusing to me as it suggests that it actually has the properties of an inner product, which it clearly cannot have.

**Limitations:**

Yes, the authors point out that kernel SVMs are usually preferred over linear SVMs and indicate that they will work on this in the future. I think that the current work is still an important step in this direction.

---

> ### Author Rebuttal · Authors · 2023-08-09
>
> # Response to Reviewer cEx8
>
> We thank the reviewer for the valuable comments and questions. Please find our responses below:
>
> > 1. geometric motivation for horospheres
>
> Our motivation behind developing a classifier based on horosphere decision boundaries was to create a large-margin classifier in Hyperbolic space that mirrors the concept of the large-margin hyperplane-based classifier (linear SVM) in Euclidean space. In Euclidean space, hyperplanes can be conceptualized as infinite-radius spheres, with parallel ones sharing the same direction.
>
> Horospheres stand out as the optimal choice for hyperbolic hyperplanes where a horosphere can be viewed as the limit of spheres as their radii approach infinity [C1]. Horospheres with the same direction are parallel to each other.
>
> Note that earlier works, such as [12,33], have defined the concept of hyperplanes in hyperbolic space using the intersection of a hyperplane in Minkowski space and the hyperboloid model. This extrinsic approach to defining the analog of Euclidean hyperplanes in hyperbolic space doesn't maintain the essential characteristic of parallel hyperplanes having the same direction.
>
>
> > 2. Figure 2 ...
>
> We used the hyperboloid SVM implementation provided by [12]. The choice of the geodesic hyperplane separator could be influenced by the optimization of the nonconvex loss function in the hyperboloid SVM. In contrast, our HoroSVM guarantees an optimal result.
>
> In the specific case in Fig.2, different choices of geodesic (by hand) might lead to fewer misclassifications, but such a choice may not be the minimizer of the distance-based loss function used in hyperboloid SVM. In our work, the choice of horospheres is quite natural as explained in A1.
>
> > 3. how the hyperbolic embeddings of the network data were generated.
>
> > 5. ...choose the optimal C
>
> We apologize for the lack of clarity in presenting details in Experiment 1. In this experiment, we followed the setup in [12] and evaluated our model on the network data embedded in hyperbolic space using the approach described in [8]. We tested 5 different embeddings of each network and report the mean and standard deviation across different embeddings for each dataset.
>
> > 4. differences are quite small
>
> The comparable performance among all methods in Table 1 for the mentioned two cases could be attributed to the fact that these datasets were well-separated. Therefore, each method is expected to perform equally well in these cases.
>
> > 6. use a Euclidean embedding
>
> The performances of Euclidean SVM on the Euclidean embedding of the data in experiment 1 are presented in [12], where Euclidean SVM performs poorly compared to hyperboloid SVM, even with high-dimensional Euclidean embeddings (d=25), e.g., for Karate, AUPR 0.5 (Euclidean) vs. 0.86 (hyperboloid SVM).
>
> > 7. embeddings in Experiment 2
>
> To present a fair comparison with the same embedding scheme, we utilize the hyperbolic entailment cone method [15] used in hyperboloid SVM [12] and hyperbolic LR [16]. We would be happy to explore the performance of all the methods using the suggested embedding scheme by de Sa et al [24] in our future work. We have to admit that we could not find any direct comparison between the two embedding methods in the suggested reference [24] as well as in any follow-up work in this domain.
>
> > 8. focus on low dimension
>
> Please refer to our response to Reviewer 38iz’s Q2
>
> > 9. parameter in Eq 2
>
> > 11. Poincare inner product
>
> We apologize for the use of the potentially misleading term, inner product. A more appropriate presentation for it will be: given an ideal point $\omega$, the function defined by $\langle \omega,x \rangle_{B}:  x \mapsto -b_{\omega}(x)$ is constant over horosphere tangent at $\omega$. Hence any horosphere can be parameterized with an ideal point $\omega$ and an offset value $b$ of the level set. As for Eq 2, $\mu$  is included for simplicity in analysis later on (Eq 3, 12). We will add a note as suggested in the revision.
>
> > 10.  in case of a collection of samples?
>
> Yes, the result holds for a collection of data. We will revise line 224-227 as follows for clarification.
>
> For a collection of training samples $S = \lbrace (x_i, y_i)\rbrace_{i=1}^N$, let $A_i =  \lbrace \nu \in \mathbb{S}^{n-1} | y_i \cdot \frac{x_i^T \nu} {\lVert x_i \rVert} > 0   \rbrace $. If the data are separable by a horosphere, it follows that $\cap_{i=1}^N A_i$ is non-empty and convex. Then the loss function given S is geodesically convex on $\mathbb{R}^{+} \times \cap_{i=1}^N A_i \times\mathbb{R}^{+}$ and the global optimum can be obtained using any gradient-based optimization.

---

> > ### Comment · Reviewer_cEx8 · 2023-08-14
> >
> > I would like to thank the authors for addressing most of my concerns and questions. However, I still have a few questions regarding the theoretical motivation for horocycles over geodesic hyperplanes.
> >
> > >
> > > 1. geometric motivation for horospheres
> >
> > If I understand it correctly, then the authors state that "horospheres stand out as the optimal choice for hyperbolic hyperplanes" due to the fact that they "maintain the essential characteristic of parallel hyperplanes having the same direction". What I am unsure of is why this property is essential. Another way to geometrically define hyperplanes is by taking the set of straight lines through some point which are orthogonal to a normal vector at that point, leading to the geodesic definition of hyperplanes. Why is this geometric definition or property inferior to the one leading to horocycles?
> >
> > >
> > > 2. Figure 2 ...
> >
> > If I understand the authors correctly, then they state that the somewhat strange choice of decision boundary in Figure 2 may be due to the (non-convexity of the) loss function. Is this a problem inherent to geodesic hyperplanes? If so, how/why? If not, then this is not an argument for horocycles over geodesic hyperplanes, but an argument for your loss function over the loss function of [12].

---

> > > ### Author Response · Authors · 2023-08-15
> > > **Response to Reviewer cEx8 #2**
> > >
> > > We thank the reviewer for the response. Please find our responses to the follow-up questions below.
> > >
> > > > 1. geometric motivation for horospheres
> > > >
> > > > (Follow-up 1) ...why this property is essential. ... Why is this geometric definition or property inferior to the one leading to horocycles?
> > > >
> > > > (Follow-up 2)...(non-convexity of the) loss function. Is this a problem inherent to geodesic hyperplanes?...
> > >
> > > In [12, 33], the hyperbolic hyperplane is defined as the intersection of the hyperboloid model and a codimension 1 subspace in Minkowski space, the ambient space of the hyperboloid model. Similarly in [11, 16], alternative definitions using the concept of Riemannian log and exponential maps also lead to geodesic hyperplanes.
> > >
> > >
> > > We choose to use horospheres as decision boundaries in hyperbolic space since they have some nice properties that were already explored in several recently published works, such as HoroPCA (Chami et al. ICML 2021), "Fully-Connected Network... "(Sonoda et al. ICML 2022) and  HyLa (Yu and De Sa. ICLR 2023). The following two geometric properties motivated our choice of horospheres as decision boundaries:
> > >
> > > (1). The important fact about parallels in hyperbolic space is that they converge onto an ideal point lying on the boundary of the Poincare disk. This concept is similar to parallels in Euclidean space meeting at an ideal point (point at infinity).
> > >
> > > (2). Further, the horospheres not only satisfy the aforementioned property but also maintain a constant hyperbolic distance between themselves. This latter property is not possessed by parallel (none intersecting) geodesic hyperplanes in hyperbolic space. This constant hyperbolic distance property facilitates the maximization of a margin that uses the concept of a "gutter" (as in Euclidean SVM [3, Sec 7.1]) which is parallel to the decision boundary (the horosphere).
> > >
> > > In addition to the above two geometric properties motivating the choice of horospheres, we also have an algebraic reason namely, our choice of horospheres as decision boundary leads to a geodesically convex loss function as proved in Theorem 3.7 in our submitted paper. In contrast, using geodesic hyperplanes leads to a non-convex loss function (see Eq 5 in [12]), primarily due to the use of the Minkowski inner product, which is an indefinite bilinear form.
> > >
> > >
> > > Reference:
> > > ***
> > > [3] Christopher M Bishop and Nasser M Nasrabadi. Pattern recognition and machine learning, volume 4 Springer, 2006.

---

### Author Rebuttal · Authors · 2023-08-09

# General response to all reviewers
We would like to thank all reviewers for their valuable comments. We are particularly encouraged that they consider the proposed method innovative (1ZfQ), beneficial for future research (cEx8),  that the theoretical result is valid (cEx8, yt6v, 1ZfQ, 38iz), the superior performance in our experiments (cEx8, yt6v, 1ZfQ, 38iz), and that our paper is well-written (yt6v) and easy to follow (1ZfQ).

We have addressed the specific questions raised by each reviewer through separate responses.  Any inaccuracies in presentation or misuse of terminology will be thoroughly clarified in the revision. We value all the reviewers' suggestions for improving the clarity and organization of our work. Modifications according to those reviews will be reflected in the revision as well. We appreciate the insightful discussion about the future direction of this research and will incorporate this feedback appropriately in the conclusion section of the revision.

We appreciate the time and effort that the reviewers spent in assessing our work. We hope that our responses have addressed all reviewers' questions and concerns. Please let us know if there are further questions.

The references cited in the rebuttal use the same indexing as in the submitted paper.

***
References:

[8] Chamberlain et al. Neural embeddings of graphs in hyperbolic space. arXiv:1705.10359, 2017.

[11] Chein et al. Highly scalable and provably accurate classification in poincaré balls. ICDM 2021.

[12] Cho et al.  Large-margin classification in hyperbolic space. AISTATS  2019.

[15] Ganea et al. Hyperbolic entailment cones for learning hierarchical embeddings. ICML 2018.

[16] Ganea et al. Hyperbolic neural networks. NeurIPS 2019.

[23] Nickel and Kiela. Poincaré embeddings for learning hierarchical representations. NeurIPS 2017.

[24] Sala et al. Representation tradeoffs for hyperbolic embeddings. ICML, 2018.

[25] Rik Sarkar. Low distortion delaunay embedding of trees in hyperbolic plane. In International Symposium on Graph Drawing, 2011.

[33] Weber et al. Robust large-margin learning in hyperbolic space. NeurIPS 2020.

[36] Zhang and Sra. First-order methods for geodesically convex optimization. COLT 2016.

***
Additional references in rebuttal:

[C1] Izumiya S. Horospherical geometry in the hyperbolic space. Noncommutativity and Singularities: Proceedings of French–Japanese symposia held at IHÉS in 2006. Mathematical Society of Japan, 2009, 55: 31-50.

[C2] Sato H. Riemannian conjugate gradient methods: General framework and specific algorithms with convergence analyses. SIAM Journal on Optimization, 2022, 32(4): 2690-2717.

[C3] Iranmehr A, Masnadi-Shirazi H, Vasconcelos N. Cost-sensitive support vector machines. Neurocomputing, 2019, 343: 50-64.

---

> ### Comment · Area_Chair_j9Sm · 2023-08-18
> **Ongoing discussions**
>
> Dear authors, reviewers,
>
> I am glad to see the back and forth between one of the reviewers and I would like to know from the other reviewers where they stand and whether they have any open points after the rebuttal (can be either a response here or below the rebuttal of your review).
>
> AC

---

### Decision · Program_Chairs · 2023-09-21

**Decision:**

Accept (poster)

**Comment:**

This paper received three ratings above the acceptance threshold and one below. The reviewer with the borderline reject rating had two concerns (experiment needed on real-world data with noise and motivation). Both were addressed by the authors. Overall, the reviewers find the HoroSVM formulation interesting, the theory solid, and the experiments in order. The AC therefore deems that the paper should be accepted.